# Speckle-Tracking Echocardiography with Novel Imaging Technique of Higher Frame Rate

**DOI:** 10.3390/jcm10102095

**Published:** 2021-05-13

**Authors:** Kana Fujikura, Mohammed Makkiya, Muhammad Farooq, Yun Xing, Wayne Humphrey, Mohammad Hashim Mustehsan, Mario J. Garcia, Cynthia C. Taub

**Affiliations:** 1Division of Cardiology, Montefiore Medical Center, Albert Einstein College of Medicine, 111 East 210th Street, Bronx, NY 10467, USA; kana.fujikura@nih.gov (K.F.); mmakkiya@montefiore.org (M.M.); mfarooq@montefiore.org (M.F.); yxing@montefiore.org (Y.X.); whumphre@montefiore.org (W.H.); mmustehs@montefiore.org (M.H.M.); mariogar@montefiore.org (M.J.G.); 2National Heart, Lung and Blood Institute, National Institutes of Health, Department of Health and Human Services, Bldg 10, Rm B1D416, 10 Center Drive, Bethesda, MD 20892, USA; 3Section of Cardiovascular Medicine, Dartmouth-Hitchcock Medical Center, 1 Medical Drive, Lebanon, NH 03766, USA

**Keywords:** echocardiography, speckle-tracking, frame rate, global longitudinal strain, left ventricle

## Abstract

**Background:** The accuracy of speckle-tracking echocardiography (STE) depends on temporal resolution. The goal of this study was to demonstrate the feasibility of relatively high frame rate (rHi-FR) (~200 fps) for STE. **Methods:** In this prospective study, echocardiographic images were acquired using clinical scanners on patients with normal left ventricular systolic function using rHi-FR and conventional frame rate (Reg-FR) (~50 FPS). GLS values were evaluated on apical 4-, 2- and 3-chamber images acquired in both rHi-FR and Reg-FR. Inter-observer and intra-observer variabilities were assessed in rHi-FR and Reg-FR. **Results:** There were 143 echocardiograms evaluated in this study. The frame rate of rHi-FR was 190 ± 25 and Reg-FR was 50 ± 3, and the heart rate was 71 ± 13. Absolute strain values measured in rHi-FR were significantly higher than those measured in Reg-FR (all *p* < 0.001). Inter-observer and intra-observer correlations were strong in both rHi-FR and Reg-FR. **Conclusions:** We demonstrated that absolute strain values were significantly higher using rHi-FR when compared with Reg-FR. It is plausible that higher temporal resolution enabled the measurement of myocardial strain at desired time point. Further investigations are necessary to evaluate the value of rHi-FR to assess myocardial strain in the setting of tachycardia.

## 1. Introduction

Echocardiography is the primary imaging modality in evaluating heart disease given its feasibility, easy accessibility, low cost and lack of ionizing radiation [1]. Global longitudinal strain (GLS) derived from speckle-tracking echocardiography (STE) detects subclinical myocardial dysfunction and can predict cardiac outcomes [2].

In STE, the pattern of ultrasound signal is tracked frame-by-frame to assess myocardial deformation during a cardiac cycle. The frame rate of conventional 2D echocardiography is approximately 40–80 fps which is usually considered adequate to evaluate myocardial deformation at normal heart rates [2,3,4]. As the accuracy of STE is highly dependent on temporal resolution, a higher frame rate than the conventional 40–80 fps may be required to obtain reliable strain values. Several techniques have been proposed in the past few years to increase frame rate in echocardiography while maintaining high image quality [5,6,7]. In a canine model, validating against myocardial strain directly measured by sonomicrometry, STE with frame rate of 211 fps reliably depicted decreased myocardial function caused by ischemia [8]. Joos P., et al. [7] demonstrated feasibility of myocardial STE at high frame rate (500 fps) in healthy volunteers using a research scanner. Recently a novel software became available to acquire echocardiographic images of the full left ventricle at a relatively high frame rate (~200 fps) (rHi-FR) using clinical ultrasound machines. In this prospective pilot study, STE was evaluated using both a standard frame rate (Reg-FR) and rHi-FR (~200 fps) on patients clinically referred for echocardiography examinations. The objective of this study was to compare the values of GLS derived from STE with rHi-FR vs. Reg-FR in a clinical setting.

## 2. Methods

### 2.1. Study Design and Populations

This was a prospective study of patients clinically referred to Adult Echocardiography Laboratory at Montefiore Medical Center between April 2017 and July 2018. Consecutive patients who agreed to undergo resting echocardiography with additional image acquisition for the purposes of the study were enrolled. The inclusion criterion was age ≥ 18 years while the exclusion criteria were significant arrhythmias and suboptimal imaging quality (defined as 2 or more suboptimal myocardial segments for strain analysis [9]). Patients who demonstrated left ventricular ejection fraction <50% were excluded from this study. The study was approved by the Office of Human Research Affairs at Albert Einstein College of Medicine (IRB# 2017-7949) and carried out according to the principles of the Declaration of Helsinki. This study contained no more than minimal risk to patients, therefore informed consent was waived by the institutional review board. 

### 2.2. Imaging Acquisition

The 2-dimensional (2D) echocardiographic images were acquired using commercially available clinical system (EPIQ 7, Philips Healthcare, Andover, MA, USA). All the images were acquired by experienced sonographers, Y.X. and W.H. Images were optimized to improve signal-to-noise ratio and provide optimal endocardial definition. Images were acquired to ensure visualization of the largest cavity lengths and with a less than 20% difference between apical 4- and 2-chamber views. Following the standard clinical protocol, additional apical 4-, 2- and 3-chamber views (e.g., the same field of view as the standard clinical protocol) were acquired with rHi-FR (~200 fps) using Hyper 2D (Philips Healthcare, Andover, MA, USA) software on the same ultrasound machine that was used to acquire clinical images. All the images were stored on digital media (IntelliSpace, Philips Healthcare, Andover, MA, USA) which is the standard protocol at Montefiore Medical Center.

### 2.3. Speckle-Tracking Echocardiography—Strain Analysis

Strain analysis was performed using Q-lab strain analysis software (CMQ, Philips Healthcare, Andover, MA, USA), a commercially available software validated for this purpose. GLS values were evaluated on apical 4-, 2- and 3-chamber images acquired in both rHi-FR and Reg-FR (Figure 1). The observer manually traced the endocardial border at end-diastole, and the software automatically tracked the border during a cardiac cycle. Adequate tracking was then visually verified, and the endocardial border was manually corrected if deemed necessary to ensure optimal tracking. After the completion of this tracking process, strain value at each frame was automatically plotted to derive a strain curve. GLS was defined as the strain at end-systole (i.e., at the time of aortic valve closure) from the three apical views (i.e., apical 4-, 2- and 3-chamber views) [2]. LV GLS curve was automatically calculated as the time-to-time average of all three apical views, and LV GLS value was derived at the end-systole. 

### 2.4. Statistical Analysis

Data were analyzed using SAS version 9.4 (Cary, NC, USA). All statistical tests were 2-tailed, and a *p*-value < 0.05 was considered statistically significant. Normality of the continuous variables was checked. Continuous variables were summarized as mean ± standard deviation based on central limit theorem. Categorical variables were presented as counts (percentages). Differences of GLS between rHi-FR and Reg-FR were calculated in apical 4-, 2-, 3-chamber images and the LV using Paired Student T-test, and the mean differences (95% confidence interval (CI)) were shown in a forest plot. Inter-observer and intra-observer variability was evaluated in 10 random rest echocardiography studies on both rHi-FR and Reg-FR sets of images [10]. For inter-observer variability, reproducibility was assessed using Pearson correlation coefficients and Bland–Altman plots. In addition, reliability was assessed by a percent change between the two observers (e.g., the absolute difference of GLS between rHi-FR and Reg-FR was divided by the mean of the repeated observations). For intra-observer variability, repeatability was assessed using Pearson correlation coefficients, and repeatability was assessed by repeatability coefficient. For assessment of reliability and reproducibility, normality of the continuous variables was checked. Continuous variables were summarized as mean ± standard deviation or median (interquartile range) depending on data distribution. 

## 3. Results

### 3.1. Patient Population and Characteristics

A total of 187 patients were recruited for this study, and 44 patients were excluded (27 for suboptimal imaging quality, 2 for arrhythmia (e.g., frequent premature ventricular contractions), and 15 for low LVEF). The final study population comprised of 143 patients. Characteristics of the cohort are summarized in Table 1.

### 3.2. GLS in rHI-FR vs. REG-FR

The frame rate of rHi-FR was 190 ± 25 and Reg-FR was 50 ± 3, and the heart rate was 71 ± 13. GLS values were compared between rHi-FR and Reg-FR in apical 4-, 2-, 3-chamber views and the LV. Absolute GLS values measured in rHi-FR were significantly higher compared to those measured in Reg-FR (all *p* < 0.001) (Figure 2).

### 3.3. Inter-Observer and Intra-Observer Variabilities

There was a strong correlation without evidence of significant bias between the measurements of the two observers in both rHi-FR (Figure 3A) and Reg-FR (Figure 3C). Bland–Altman plots showed minimal bias between measurements and narrow limits of agreement between the two observers in both rHi-FR (Figure 3B) and Reg-FR (Figure 3D). The reliability calculated as a percent change in two observers was 6.7 [4.4, 7.4] % (*p* < 0.0001) in rHi-FR and 5.5 [4.0, 7.8] % (*p* < 0.0001) in Reg-FR. 

Intra-observer variability assessment showed strong correlation between the two measurements in both rHi-FR (r = 0.98, regression 0.98 95% CI (0.85, 0.98), ICC 0.98, *p* < 0.0001) (Figure 4A) and Reg-FR (r = 0.95, regression 1.02 95% CI (0.89, 1.16), ICC 0.96, *p* < 0.0001) (Figure 4B). Repeatability coefficient of rHi-FR was 1.7% 95% CI (0.83%, 2.49%), and Reg-FR was 2.4% 95% CI (1.11%, 2.77%).

## 4. Discussion

STE has been increasingly adopted for the assessment of LV function in various clinical settings. STE is highly dependent on temporal resolution, and our result indicates that rHi-FR (~200 fps) provided higher absolute values than Reg-HR STE (~50 fps). Optimizing GLS measurements is clinically important because GLS is proven to be useful in detecting subclinical LV mechanical changes. The ability of rHi-FR STE to detect true GLS in patients with normal LVEF is potentially powerful and meaningful in a clinical setting.

Using a novel software (e.g., Hyper 2D) on a standard clinical ultrasound machine, 2D echocardiography images can be obtained with rHi-FR (~200 fps) which is more than twice that of the conventional frame rate. Hyper 2D uses divergent transmit beams in order to cover the same region with fewer transmits. Divergent beams have a negative focus such that the beams are wide, allowing for a faster frame rate. In Hyper 2D, divergent beams are used in combination with high-order receive multiline. The image is formed by storing the received acoustic signals from the multiple transmit events. Received lines in the same location from the multiple transmit beams can be combined to produce an image with improved lateral resolution. The intent of the Hyper 2D scanning method is to achieve much higher frame rates with nearly the same resolution as the traditional 2D scanning method. To our knowledge, this is the first paper evaluating LV strain using images acquired with Hyper 2D.

Historically, tissue Doppler echocardiography was used to evaluate ventricular function by strain analysis to circumvent limited temporal resolution [11]. However, Doppler technique is based on unidirectional ultrasound beam, and thus it tracks myocardial motion along the direction of ultrasound beam. The frame rate of Doppler-based strain assessment is >180 fps, and recently it has been even increased to ~500 fps. In order to achieve high frame rate, lateral resolution has been sacrificed by increasing line spacing and the beam width. On the other hand, 2D speckle-tracking technique is angle-independent, and it is sensitive to lateral motion. However, frame rate with 2D image is traditionally lower (e.g., 40–80 fps) than that of Doppler. The performance of 2D speckle-tracking is dependent on both the spatial and temporal resolution. Speckle-tracking is considered to improve at higher frame rate due to lower speckle decorrelation between frames. Therefore, strain analysis by 2D STE using ultra-fast frame rate echocardiography has been studied for over a decade [12]. Lee et al. demonstrated the feasibility of STE in a canine model to detect ischemic myocardium using STE with similar frame rate to this study by comparing strain evaluated with STE to sonomicrometry [8]. Sonomicrometry is an in-vivo method that directly measures myocardial strain using a paired crystals implanted in the myocardium. Hyper 2D is a novel technology that acquires 2D echocardiographic images of the full left ventricle in relatively high frame using clinical ultrasound machines. This technique may be promising because it can be easily applied in our daily clinical practice to provide quality strain analysis. 

Our results show significantly higher GLS with rHi-FR compared to Reg-FR in patients with normal LVEF. It is plausible that higher temporal resolution (e.g., higher frame rate) enabled the measurement of myocardial strain at close to end-systole [13,14] as illustrated in Figure 5. Using cine imaging with conventional frame rate, reproducibility of strain values has been verified to be sufficient for clinical use [15]. However, it is important to recognize that reproducibility is to show precision but not necessarily accuracy of the measurements. Therefore, strain values evaluated in rHi-FR may be closer to the true end-systolic GLS (e.g., GLS at the ideal point of end-systole) compared to the values evaluated in Reg-FR.

Furthermore, rHi-FR echocardiography may expand STE application to higher frame rate condition such as stress echocardiography. As stated above, it is generally considered that frame of 40 to 80 fps is relatively low and only adequate for resting heart rate. In case of increased heart rate, there is a potential of underestimating the peak strain value [16]. Previous data on normal myocardial strain response to exercise are somewhat contradictory and limited. In stress echocardiography, the accuracy of detecting significant coronary artery disease (CAD) was reported as 60% using STE with 60–100 fps [17]. Joyce, et al. [17] also showed that the GLS at peak stress was smaller than GLS at rest in all their cohort regardless of the presence of significant CAD. Yu Y., et al. [18] also showed a marked decline in strain in individuals without obstructive CAD with dobutamine stress test especially at low doses (−17.6% to −10.8%). Sanchez A.A., et al. [19] demonstrated that the ratio of frame rate to heart rate >0.7 frames/sec per bpm is sufficient for reproducible strain value using a premature infant of 26 ± 1 weeks with the heart rate of 163 ± 13 bpm. RHi-FR has the potential to solve these problems, therefore extending the utility of STE to stress echocardiography in the future.

Our study showed excellent inter-observer reproducibility and agreement of GLS in both rHi-FR and Reg FR by Pearson correlation, linear regression, intraclass correlation coefficient, bias, and Bland–Altman plots. The inter-observer reliability test showed significant difference between the two observers; however, the actual number of percent change between the two observers was less than 8% that was thought to be reliable. Repeatability coefficient of two measurements by one observer was reasonable in both rHi-FR and Reg-FR. We used a strain analysis software from a single vendor (CMQ, Philips Healthcare) for all the STE analysis, and our observations were consistent with the results reported by EACVI/ASE/Industry Task Force to Standardize Deformation Imaging [20]. 

### Study Limitations

This study has some limitations. First, there was a possibility of selection bias because patients with suboptimal image quality were excluded from this study. Furthermore, evaluation using suboptimal image quality would result in unreliable results. Therefore, exclusion due to suboptimal image quality is a common limitation of cardiovascular imaging studies in general. Secondly, this pilot study only included patients with normal LVEF. Additionally, STE was evaluated only at rest. Future studies are needed to evaluate the utility of rHi-FR in patients with reduced LVEF and also in conditions with increased heart rates. Thirdly, echocardiography images were acquired using echocardiography machines of a single vendor. STE was analyzed using a software of a single vendor as well. Similar studies need to be performed using echocardiography machines and STE analyzing software of different vendors to verify that our results are neutral to vendors.

## 5. Conclusions

We demonstrated that absolute strain values were significantly higher using rHi-FR when compared with Reg-FR in patients with normal LVEF. It is plausible that higher temporal resolution enabled the measurement of strain at the optimal timing. The result of this study may inform clinical adoption of the novel technology. Further investigations are necessary to evaluate the value of rHi-FR to assess myocardial strain in the setting of increased heart rate.

## Figures and Tables

**Figure 1 jcm-10-02095-f001:**
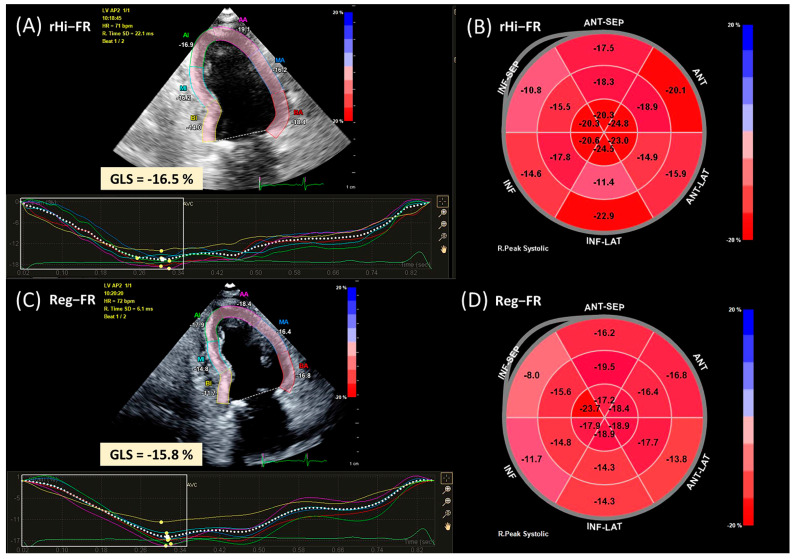
Samples of speckle-tracking analysis. Strain values were evaluated in (**A**,**B**) rHi-FR and (**C**,**D**) Reg-FR. rHi-FR, relatively high frame rate; Reg-FR, regular frame rate.

**Figure 2 jcm-10-02095-f002:**
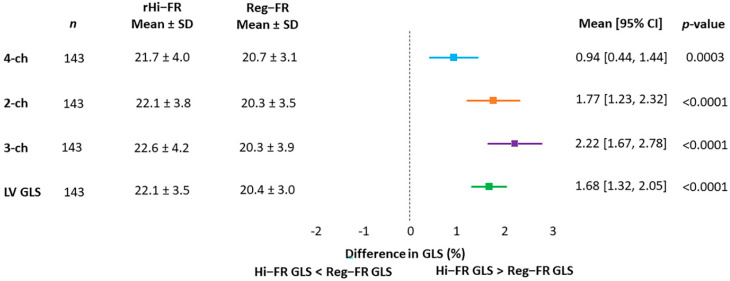
Forest plot showing difference in absolute GLS values measured in rHi-FR vs. Reg-FR. GLS, global longitudinal strain; rHi-FR, relatively high frame rate; LVEF, left ventricular ejection fraction; Reg-FR, regular frame rate.

**Figure 3 jcm-10-02095-f003:**
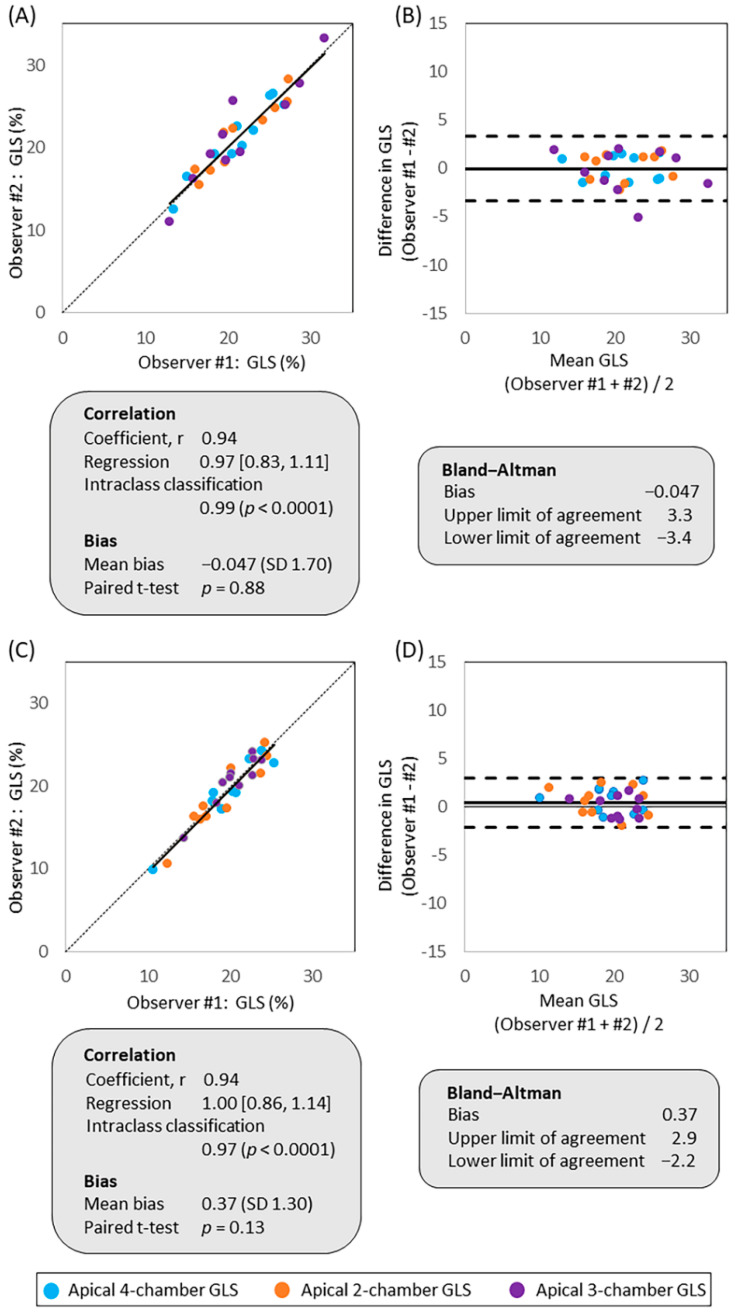
Inter-observer variability in apical 4-, 2-, and 3-chamber views. Reproducibility and agreement of absolute GLS values measured by observer #1 and #2 were compared using linear regression and Bland–Altman plots in (**A**,**B**) rHi-FR and (**C**,**D**) Reg-FR. GLS, global longitudinal strain; rHi-FR, relatively high frame rate; Reg-FR, regular frame rate.

**Figure 4 jcm-10-02095-f004:**
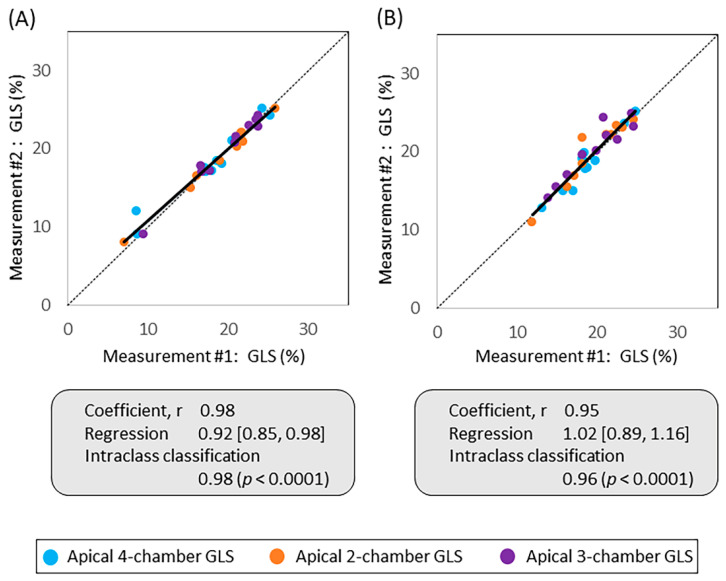
Intra-observer variability in apical 4-, 2-, and 3-chamber views. Linear regression comparing absolute GLS values of 1st and 2nd measurement in (**A**) rHi-FR and (**B**) Reg-FR. GLS, global longitudinal strain; rHi-FR, relatively high frame rate; Reg-FR, regular frame rate.

**Figure 5 jcm-10-02095-f005:**
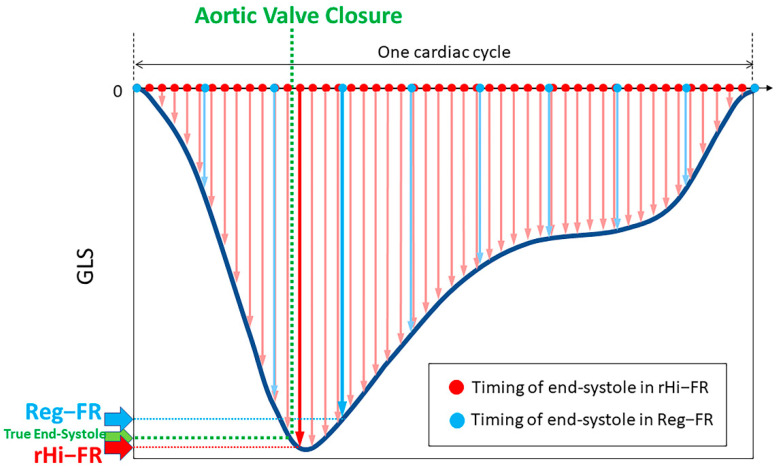
A schematic example of GLS values in rHi-FR and Reg-FR. The dark blue line represents an optimal GLS curve. In this schematic example, the GLS value in rHi-FR is larger and closer to the value at true end-systole compared with the value in Reg-FR. GLS, global longitudinal strain; rHi-FR, relatively high frame rate; Reg-FR, regular frame rate.

**Table 1 jcm-10-02095-t001:** Patient characteristics.

	*n* = 143
Male, *n* (%)	47 (32.9)
Age, mean ± SD (y)	60.1 ± 15.1
Ethnicity	
African American, *n* (%)	42 (29.4)
Hispanic, *n* (%)	61 (42.7)
Caucasian, *n* (%)	10 (7.0)
Asian, *n* (%)	2 (1.4)
Others, *n* (%)	28 (19.6)
Hypertension, *n* (%)	96 (67.1)
Diabetes, *n* (%)	42 (29.4)
Coronary artery disease, *n* (%)	25 (17.5)
Hyperlipidemia, *n* (%)	74 (51.8)
Body surface area, mean ± SD (m^2^)	1.84 ± 0.23
Systolic blood pressure, mean ± SD (mmHg)	133.3 ± 23.2
Diastolic blood pressure, mean ± SD (mmHg)	77.2 ± 11.8
Heart rate, mean ± SD (beat per minute)	70.6 ± 12.7
LVEF, mean ± SD (%)	63.0 ± 5.7

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
