# Peer review of "Speckle-Tracking Echocardiography with Novel Imaging Technique of Higher Frame Rate"

_jcm, 2021, doi:10.3390/jcm10102095_

Round 1

Reviewer 1 Report

The study of Dr Fujikura&Dr. Makkiya presents a new technological approach used for high(er) frame rate STE in normal LV function patients at rest. This is interesting and may have relevance for clinical practice. I have however some comments:

  1. Hyper 2D is already used for 3D applications. In order to make your work understandable you should explain a little how this works on the images, what is the difference with normal imaging (lesser transmits), and what other work was published with your method (if any)
  2. 200 Hz is not high frame rate. It is high(er) frame rate. Previous Philips systems were capable of delivering small-window frame rates of 200 Hz, and speckle-tracking algorithms were already used for these images, albeit not for the GLS. 2D data extracted from Philips clinical TDI was already analyzed at 500 Hz FR
  3. The discussion is highly speculative, the analysis lacks reference. The results are interesting, but you should only state what can be demonstrated.
  4. The Discussion covers some statistical and physiological points. This is a new approach. I would like to see in the discussion more about the underlying technical ground and justification on why and how this method would be superior. I would like to have the demonstration that you deeply understand the advantages and drawbacks of high(er) frame rate

Other points as they arise during lecture:

Title: while it is novel, it is questionable if high frame rate really applies to 200Hz.

Results:

  • PVC’s are not severe arrhythmia

Table 1:

Interesting that the population comprised only 7% Caucasian, is that because of the patient population in your center?  

Figure legends:

In the legend you describe panel C, in both figures, but you skip B. Probably you mean B.

General: you should refer to higher values as higher absolute values, since strain is negative in systole, and scientifically speaking higher is less negative.

Discussion:

  • You argue that the ability to detect true GLS is important. How do you know which is the “true ” GLS?
  • “Frame rate cannot be modified during post-processing to analyse STE”. I do not get the meaning of this statement. Would anyone expect to modify frame rate (increase) in postprocessing? Stricto sensu you can modify the frame rate in postprocessing by decimating or skipping frames. But you cannot increase the quantity of frames/information.
  • You state that higher values suggest more accurate strain by HFR. Without reference I could only state that there is a positive bias with the HFR method.

  • Figure 5 is a theoretical figure, I do not understand how this figure proves what you state. It is a hypothesis, a speculation, but no facts support the truth in it.
  • The problem of stress STE is not only limited to frame rate. Image quality is also decreased. And it is expected also to be decreased in HFR echo, so I am not entirely sure about the better results of HFR echo in stress STE.
  • Also the decrease in GLS with tachycardia may not be solely due to frame rate. The strain is the relative deformation, which may be less during effort just because of lesser filling/distension, in spite of faster and better emptying. We are talking about linear strain for a 3D deformation
  • You state that your study showed great reproducibility. Aside the fact that “great” is not scientifically balanced, when talking about one’s work, linear correlation is not the best way to demonstrate variability, especially while analyzing subjects with normal function and normal strain, thus tightly packed values around the normal.
  • You state that your study gave results consistent with the results of the taskforce on strain. You conclude that this means your study is precise and reliable. I do not agree with the logic in this statement.

Limitations:

  • I do not understand the paragraph on arrhythmia and image quality. Can be deleted

Figure 1: GLS values are a little on the abnormal side for both HFR and normal FR.

Author Response

We appreciate your detailed and insightful comments. Below please find our responses.

  1. Hyper 2D is already used for 3D applications. In order to make your work understandable you should explain a little how this works on the images, what is the difference with normal imaging (lesser transmits), and what other work was published with your method (if any)

Thank you for your comment on this important point.

The following explanation is added in the second paragraph of our Discussion section.

“Hyper 2D uses divergent transmit beams in order to cover the same region with fewer transmits. Divergent beams have a negative focus such that the beams are wide, allowing for a faster frame rate. In Hyper 2D, divergent beams are used in combination with high order receive multiline. The image is formed by storing the received acoustic signals from the multiple transmit events. Received lines in the same location from the multiple transmit beams can be combined to produce an image with improved lateral resolution. The intent of the Hyper 2D scanning method is to achieve much higher frame rates with nearly the same resolution as the traditional 2D scanning method. Our study is the first paper evaluating LV strain using images acquired with Hyper 2D.

Hyper 2D is indeed the basis for current fast 3D imaging techniques as each elevation plane in a 3D volume is formed using the same technique. However, the frame rate of current fast 3D for the entire left ventricle is approximately 30 Hz which is significantly lower than that of Hyper 2D.

To our knowledge, this manuscript will be the first publication of strain analysis using Hyper 2D. This point is now clarified in the last sentence of the second paragraph of our Discussion section.

  1. 200 Hz is not high frame rate. It is high(er) frame rate. Previous Philips systems were capable of delivering small-window frame rates of 200 Hz, and speckle-tracking algorithms were already used for these images, albeit not for the GLS. 2D data extracted from Philips clinical TDI was already analyzed at 500 Hz FR

While a 200 Hz frame rate is not high, as all things are, relative. The text of our manuscript states that it is relatively high frame rate. In order to clarify this point, the abbreviation ‘Hi-FR’ is now changed to ‘rHi-FR’. Additionally, we revised our title to reflect this point. The revised title is “Speckle-tracking echocardiography with novel imaging technique of higher frame-rate”.

Previous Philips systems could indeed achieve ~200 Hz when a field of view (FOV) was set to be very small with shallow depth. This FOV was not feasible to assess the entire left ventricle (LV). The purpose of this study was to evaluate feasibility of Hyper 2D to assess myocardial deformation of the entire LV.

TDI-based strain can indeed achieve high frame rate. In order to achieve higher frame rates with TDI-based strain, the lateral resolution was sacrificed. This was done by increasing the line spacing and increasing the beam width. Hyper 2D allows speckle-tracking strain analysis that is different from TDI strain analysis. TDI have a very low lateral resolution, whereas Hyper 2D has the same lateral resolution as that of a 2D image.

  1. The discussion is highly speculative, the analysis lacks reference. The results are interesting, but you should only state what can be demonstrated.

We agree with the reviewer’s comment that our statement is an assumption based on scientific knowledge. In the last sentence of the 4th paragraph of our Discussion section, ‘are’ is now changed to ‘may be’.

Of note, relation between frame rate and accuracy of strain value has been evaluated in-vivo using sinusoidal compressions*. The influence of temporal resolution on strain measurement is also illustrated in a recent review paper published in European Heart Journal - Cardiovascular Imaging†. The reference papers are now added to our Discussion as reference #13(*) and #14(†) to support our statement of accuracy in GLS values measured with rHi-FR.

In addition, reference #15() is now added to clarify a prior study that showed reproducibility of strain values using conventional frame rate.

* Chen H, et al. Ultrasound frame rate requirements for cardiac elastography: Experimental and in vivo results. Ultrasounics 2008;49:98-111

† Amzulescu MS, et al. Myocardial strain imaging: review of general principles, validation, and sources of discrepancies. Eur Heart J Cardiovasc Imaging. 2019;20:605–619.

Cheng S, et al. Reproducibility of speckle-tracking-based strain measures of left ventricular function in a community-based study. J Am Soc Echocardiogr. 2013;26:1258–66.

  1. The Discussion covers some statistical and physiological points. This is a new approach. I would like to see in the discussion more about the underlying technical ground and justification on why and how this method would be superior. I would like to have the demonstration that you deeply understand the advantages and drawbacks of high(er) frame rate

Thank you for your request. Now the Discussion section is modified. The 2nd paragraph of the Discussion has been revised, the 3rd paragraph has been deleted, and the original 4th paragraph (now 3rd paragraph) has been revised. The revised 2nd  and 3rd paragraph are as follows.

“      Using a novel software (e.g. Hyper 2D) on a standard clinical ultrasound machine, 2D echocardiography images can be obtained with rHi-FR (~200fps) which is more than twice that of the conventional frame rate. Hyper 2D uses divergent transmit beams in order to cover the same region with fewer transmits. Divergent beams have a negative focus such that the beams are wide, allowing for a faster frame rate. In Hyper 2D, divergent beams are used in combination with high order receive multiline. The image is formed by storing the received acoustic signals from the multiple transmit events. Received lines in the same location from the multiple transmit beams can be combined to produce an image with improved lateral resolution. The intent of the Hyper 2D scanning method is to achieve much higher frame rates with nearly the same resolution as the traditional 2D scanning method. Our study is the first paper evaluating LV strain using images acquired with Hyper 2D.

        Historically, tissue Doppler echocardiography was used evaluate ventricular function by strain analysis to circumvent limited temporal resolution. However, Doppler technique is based on an unidirectional ultrasound beam, and so it tracks myocardial motion along the direction of ultrasound beam. The frame rate of Doppler-based strain assessment is >180 fps, and recently it has been even increased to ~500 fps. In order to achieve high frame rate, lateral resolution has been sacrificed by increasing line spacing and the beam width. On the other hand, 2D speckle-tracking technique is angle-independent and it is sensitive to lateral motion. However, frame rate with 2D image is traditionally lower (e.g. 40-80 fps) than that of Doppler. The performance of 2D speckle-tracking is dependent on both the spatial and temporal resolution. Speckle-tracking is consider to improve at higher frame rate due to lower speckle decorrelation between frames. Therefore, strain analysis by 2D STE using ultra-fast frame rate echocardiography has been studied for over a decade. Lee et al. demonstrated the feasibility of STE in canine model to detect ischemic myocardium using STE with similar frame rate to this study by comparing strain evaluated with STE to sonomicrometry. Sonomicrometry is an in-vivo method that directly measures myocardial strain using a paired crystals implanted in the myocardium. Hyper 2D is a novel technology that acquires 2D echocardiographic images in relatively high frame using clinical ultrasound machines. This technique is may be promising because it can be easily applied in our daily clinical practice to provide quality strain analysis.”

Other points as they arise during lecture:

Title: while it is novel, it is questionable if high frame rate really applies to 200Hz.

We agree with your comment. We’ve modified our title to reflect this point. The new title is “Speckle-tracking echocardiography with novel imaging technique of higher frame-rate”.

Results:

  • PVC’s are not severe arrhythmia

We agree with this statement and now the word ‘severe’ is removed.

Table 1:

Interesting that the population comprised only 7% Caucasian, is that because of the patient population in your center?  

Yes. The majority of patients at our medical center is Hispanic and African American.

Figure legends:

In the legend you describe panel C, in both figures, but you skip B. Probably you mean B.

Thank you for noticing our typos. Panel C was meant to be panel B. It is now corrected in both Figure 3 and Figure 4.

General: you should refer to higher values as higher absolute values, since strain is negative in systole, and scientifically speaking higher is less negative.

We agree with the reviewer’s comment. Now the word ‘absolute’ is added to describe strain values that were absolute values.

Discussion:

  • You argue that the ability to detect true GLS is important. How do you know which is the “true ” GLS?

As stated in #3 above, higher frame rate has been shown to capture more accurate strain value using in-vivo sinusoidal compressions*.  The influence of temporal resolution on strain measurement is also illustrated in a recent review paper published in European Heart Journal - Cardiovascular Imaging†. The references are now added as reference #13(*) and #14(†) to clarify that our statement is based on previous publications.

* Chen H, et al. Ultrasound frame rate requirements for cardiac elastography: Experimental and in vivo results. Ultrasounics 2008;49:98-111.

† Amzulescu MS, et al. Myocardial strain imaging: review of general principles, validation, and sources of discrepancies. Eur Heart J Cardiovasc Imaging. 2019;20:605–619.

  • “Frame rate cannot be modified during post-processing to analyse STE”. I do not get the meaning of this statement. Would anyone expect to modify frame rate (increase) in postprocessing? Stricto sensu you can modify the frame rate in postprocessing by decimating or skipping frames. But you cannot increase the quantity of frames/information.

We agree that the sentence is confusing, and so we removed it from our revised manuscript.

  • You state that higher values suggest more accurate strain by HFR. Without reference I could only state that there is a positive bias with the HFR method.

Thank you for your comment on this great point. We agree that higher values suggest more accurate strain without reference values. However, based on the prior in vivo study by Chen H*, et al, it can be said that “it is plausible that higher temporal resolution (e.g. higher frame rate)  enabled the measurement of myocardial strain at close to end-systole”.

* Chen H, et al. Ultrasound frame rate requirements for cardiac elastography: Experimental and in vivo results. Ultrasounics 2008;49:98-111.

  • Figure 5 is a theoretical figure, I do not understand how this figure proves what you state. It is a hypothesis, a speculation, but no facts support the truth in it.

Figure 5 illustrates the concept of the timing of true end-systole, rHi-FR, and Reg-FR as an example. The relation of strain values at these three points may vary depending on the strain curve and the timing of those three points in individual image. To clarify this point, the figure legend of Figure 5 is now modified as follows.

“A schematic example of GLS values in rHi-FR and Reg-FR. The dark blue line represents an optimal GLS curve. In this schematic example, the GLS value in rHi-FR is larger and closer to the value at true end-systole compared with the value in Reg-FR.”

  • The problem of stress STE is not only limited to frame rate. Image quality is also decreased. And it is expected also to be decreased in HFR echo, so I am not entirely sure about the better results of HFR echo in stress STE.

We agree with the reviewer’s statement. In general, the image quality of stress echocardiogram is less optimal compared to rest echocardiography, with the exception of pharmacologic stress modality. However, the amount of decrease in image quality is unknown. In addition, as stated in our answer to your comment #1, the intent of the Hyper 2D scanning method is to achieve much higher frame rates with nearly the same resolution as the traditional 2D scanning method. It is of our interest to evaluate strain to evaluate the feasibility of Hyper 2D in increased heart rate induced by stress.  

  • Also the decrease in GLS with tachycardia may not be solely due to frame rate. The strain is the relative deformation, which may be less during effort just because of lesser filling/distension, in spite of faster and better emptying. We are talking about linear strain for a 3D deformation

We agree your statement. The decrease in GLS is probably multifactorial. As you stated, we are evaluating 3D deformation using 2D technology. GLS can be affected by various physiological conditions (e.g. intracavitary pressure and volume) in addition to under-sampling associated with limited temporal resolution especially when the heart rate is increased. However, myocardial strain under the condition of increased heart rate has been studied in premature infants (26 ± 1 weeks), and it has been demonstrated that reproducible strain values were obtained when FR/HR ration> 0.7 frames/sec per bpmǁ. In this study, the heart rate was 163 ± 13 bpm, and cine images were acquired at frame rate of <90, 90-110, and > 130 Hz.

ǁ Sanchez AA, et al. Effects of frame rate on two-dimensional speckle tracking-derived measurements of myocardial deformation in premature infants. Echocardiography 2015;32:839-847.

  • You state that your study showed great reproducibility. Aside the fact that “great” is not scientifically balanced, when talking about one’s work, linear correlation is not the best way to demonstrate variability, especially while analyzing subjects with normal function and normal strain, thus tightly packed values around the normal.

Thank you for your comment. We agree that the word “great” is misleading. Now ‘great reproducibility’ is changed to ‘good reproducibility’. Inter- and intra-observer variabilities were evaluated in 10 random echo studies. The range of absolute strain value was widely spread. For inter-observer variability, the range of absolute strain values were approximately 12 – 33% in Hi-FR and 10 – 25% in Reg-FR. For inter-observer variability, the range of absolute strain values were approximately 8 – 25%. Therefore we think linear correlation is a valid approach to assess reproducibility.

  • You state that your study gave results consistent with the results of the taskforce on strain. You conclude that this means your study is precise and reliable. I do not agree with the logic in this statement.

We agree the conclusion can be misleading. The sentence is now removed.

Limitations:

  • I do not understand the paragraph on arrhythmia and image quality. Can be deleted

We agree with the reviewer’s suggestion. Now the statements regarding arrhythmia have been removed from the Limitation section.

Figure 1: GLS values are a little on the abnormal side for both HFR and normal FR.

We agree that the GLS values of the particular example on Figure 1 are on the abnormal side.

Reviewer 2 Report

Quite interesting  report. I have same question

1. 17,5% subjects (25pts) had coronary artery disease. In my opinion, they should be excluded from that group or GLS assessed by HI-FR and reg_FR should be discussed,  could you  comment if they had the same and the correct GLS in both methods?.

Did you notice any relation between heart rate and GLS by HiFR and were there  any pts with LBBB or LAH?

Author Response

Thank you for your comments. Please find our answers beneath each of your statement.

  1. 17,5% subjects (25pts) had coronary artery disease. In my opinion, they should be excluded from that group or GLS assessed by HI-FR and reg_FR should be discussed,  could you  comment if they had the same and the correct GLS in both methods?.

Thank you for your suggestion. We obtained both rHi-FR and Reg-FR images from each patient per echo study, and performed paired analysis. Therefore, we do not think regional wall motion abnormalities associated with coronary artery disease have affected our result. The purpose of this study was to evaluate the feasibility of Hi-FR in GLS assessment. Therefore, we did not assess the effect of coronary artery disease on GLS analysis.

Did you notice any relation between heart rate and GLS by HiFR and were there  any pts with LBBB or LAH?

Thank you for your comment. This is a pilot study to assess the feasibility of Hi-FR in strain analysis. Therefore, we did not assess the effect of heart rate or conduction abnormalities (e.g. LBBB or LAH) on GLS. As you mentioned, heart rate or conduction abnormalities are important parameters that may skew GLS values. These parameters would be evaluated as subsequent studies.

Round 2

Reviewer 1 Report

I would like to thank the authors for their efforts on improving their paper. It is consistently improved and the elements that still raise questions are less important for the general message.

The points that persist:

Methods:

  • The point on the full LV in the image is probably important to note somewhere in the beginning, in order to explain why 200 hz is indeed very significant.
  • The answer concerning the HR-to FR ratio should be somewhere in the discussion, at the point about effort GLS.

The point on inter/intra-observer analysis: a correlation between two observations does not show reproducibility, but only correlation, because it does not account for bias. If you prefer to keep the correlation analysis as method of interobserver variability, then the correlation of data should only demonstrate correlation not reproducibility. Further, for interobserver analysis, reproducibility cannot be assessed by correlation. For the intraobserver the strictly correct term is repeatability, because measurement conditions are the same. But reproducibility and repeatability refer to the same analysis, one with changing and other without changing conditions. So:

  • Intraobserver analysis of agreement used repeatability coefficient, it should also be given with CI, to give an idea of how precisely it has been estimated. The reliability has not been assessed, but ICC should be used (with CI). In fact you used correlation and repeatability coefficient which address the same agreement question.
  • For interobserver variability the Pearson correlation does not confirm reproducibility, but gives an idea on correlation, while bias is not accounted for. A true estimate of reproducibility would be for example a Bland-Altman test. Further, reliability might be estimated by ICC with CI of course ( ICC is an estimate, so CI are necessary to understand its meaning).
